# Observational study of agreement between attending and trainee physicians on the surprise question: "Would you be surprised if this patient died in the next 12 months?"

Christopher J. Yarnell[1,2,3]*, Laura M. Jewell[4], Alex Astell[5], Ruxandra Pinto[6], Luke A. Devine[2,7], Michael E. Detsky[2,3], James Downar[8,9], Roy Ilan[10], Shail Rawal[7,11], Natalie Wong[3,7,12], John J. You[13], Rob A. Fowler[1,3,6,14]

1 Institute of Health Management, Policy, and Evaluation, University of Toronto, Toronto, Canada, 2 Department of Medicine, Sinai Health System, Toronto, Canada, 3 Interdepartmental Division of Critical Care Medicine, University of Toronto, Toronto, Canada, 4 Memorial University of Newfoundland, Discipline of Family Medicine, Happy Valley-Goose Bay, Canada, 5 University of Manitoba Faculty of Medicine, Section of Critical Care Medicine, Manitoba, Canada, 6 Sunnybrook Health Sciences Centre Department of Critical Care, Toronto, Canada, 7 University of Toronto Temerty Faculty of Medicine, Division of General Internal Medicine, Toronto, Canada, 8 The Ottawa Hospital, Ottawa, Canada, 9 University of Ottawa Faculty of Medicine, Division of Palliative Care, Ottawa, Canada, 10 Department of Critical Care Medicine, Rambam Health Care Campus, Technion, Israel Institute of Technology, Haifa, Israel, 11 University Health Network, General Internal Medicine, Toronto, Canada, 12 Departments of General Internal Medicine and Critical Care Medicine, St Michael's Hospital, Toronto, Canada, 13 Division of General Internal and Hospitalist Medicine, Department of Medicine, Trillium Health Partners, Mississauga, Ontario, Canada, 14 Institute for Clinical Evaluative Sciences, Toronto, Canada

* christopher.yarnell@sinaihealth.ca

## Abstract

### Background

Optimal end-of-life care requires identifying patients that are near the end of life. The extent to which attending physicians and trainee physicians agree on the prognoses of their patients is unknown. We investigated agreement between attending and trainee physician on the surprise question: "Would you be surprised if this patient died in the next 12 months?", a question intended to assess mortality risk and unmet palliative care needs.

### Methods

This was a multicentre prospective cohort study of general internal medicine patients at 7 tertiary academic hospitals in Ontario, Canada. General internal medicine attending and senior trainee physician dyads were asked the surprise question for each of the patients for whom they were responsible. Surprise question response agreement was quantified by Cohen's kappa using Bayesian multilevel modeling to account for clustering by physician dyad. Mortality was recorded at 12 months.

### Results

Surprise question responses encompassed 546 patients from 30 attending-trainee physician dyads on academic general internal medicine teams at 7 tertiary academic hospitals in

**Data Availability Statement:** All relevant deidentified data are within the manuscript and its Supporting Information files.

**Funding:** The study was funded by a Resident Research Grant from the Physician Services Incorporated Foundation (Dr Yarnell, R15-40). Dr Yarnell is also funded by the Canadian Institutes for Health Research CGS-M, the Clinician Investigator Program, and the Eliot Phillipson Clinician Scientist Training Program at the University of Toronto. The opinions, results and conclusions reported in this paper are those of the authors and are independent from the funding sources including the PSI Foundation. No endorsement by any of the funding agencies is intended or should be inferred. The funders had no role in study design, data collection and analysis, decision to publish, or preparation of the manuscript.

**Competing interests:** The authors have declared that no competing interests exist.

Ontario, Canada. Patients had median age 75 years (IQR 60–85), 260 (48%) were female, and 138 (25%) were dependent for some or all activities of daily living. Trainee and attending physician responses agreed in 406 (75%) patients with adjusted Cohen's kappa of 0.54 (95% credible interval 0.41 to 0.66). Vital status was confirmed for 417 (76%) patients of whom 160 (38% of 417) had died. Using a response of "No" to predict 12-month mortality had positive likelihood ratios of 1.84 (95% CrI 1.55 to 2.22, trainee physicians) and 1.51 (95% CrI 1.30 to 1.72, attending physicians), and negative likelihood ratios of 0.31 (95% CrI 0.17 to 0.48, trainee physicians) and 0.25 (95% CrI 0.10 to 0.46, attending physicians).

## Conclusion

Trainee and attending physician responses to the surprise question agreed in 54% of cases after correcting for chance agreement. Physicians had similar discriminative accuracy; both groups had better accuracy predicting which patients would survive as opposed to which patients would die. Different opinions of a patient's prognosis may contribute to confusion for patients and missed opportunities for engagement with palliative care services.

## Introduction

Provision of optimal end-of-life care requires reliable identification of patients likely to die in the near future. Unfortunately, few reliable bedside tools are available to identify patients who may benefit from goals-of-care discussions due to near-term risk of death [1,2]. Predictive models based on illness severity are intended for use at the population level and may be misleading when applied to individual patients [1,3–5]. While some patients suffer from terminal diseases with predictable trajectories, many suffer from multiple comorbidities conferring an unpredictable trajectory towards death [6]. Uncertainty in patient prognosis can increase patient distress, impair communication between physicians and patients, and delay appropriate adoption of a palliative approach [7]. Further, the risk of death may not correlate with other important aspects of patient care such as uncontrolled symptoms or psychological distress [8,9]. Clinician estimates of a patient's risk of death are fallible, although accuracy may correlate with clinician experience, and these estimates likely have a large influence on end-of-life care planning [8,10–15].

One way to gain insight into clinician estimates of prognosis is to investigate how prognostic accuracy varies with training by comparing the prognostic estimates of attending and trainee physicians. Studying differences in prognostic estimates between trainee and attending physicians provides a way to investigate the development of prognostic skills, including whether or not accuracy improves with clinical training. Prognostic discordance and covariates associated with discordance may identify an opportunity for education by highlighting the patient characteristics that make prognosis more difficult. Discordance between trainee and attending physician prognoses also has practical implications because trainee physicians play an important role for inpatients at academic health centres, where discordance may cause confusion in clinical plans, mixed messages for patients, and may partially explain observed differences between received and documented patient end-of-life care preferences [16,17]. The prevalence of discordance between attending and trainee physician prognoses is unknown.

The surprise question is a simple tool to assess patient prognosis and screen for patients who may benefit from end-of-life care [18–21]. The surprise question asks "Would you be

surprised if this patient died in the next 12 months?" and a response of "No" is intended to identify a patient with potentially unmet palliative care needs [22]. In the absence of a gold standard for unmet palliative care needs, most evaluation of the surprise question has instead focused on mortality. A "no" response is correlated with increased mortality rates and discriminatory ability is similar to other prognostic indices intended for hospitalized older adults [1,2,23,24]. Hospitalized patients may be at higher risk of unmet palliative care needs and mortality because of the situation leading to admission and potential gaps in the social safety net that may precipitate hospital admission [25–27]. Improving the identification of patients at high risk of dying can improve clinical care through providing more certainty to patients, more clarity for clincians making recommendations about invasive interventions or investigations, and triggering earlier appropriate activation of palliative care services. Therefore we compared attending and trainee physician responses to the surprise question in a general internal medicine population to assess the extent and prevalence of discordance as well as the prognostic value of the surprise question with respect to mortality in this population.

## Methods

### Study population

This was a prospective cohort study of general internal medicine inpatients and their corresponding physicians at 7 tertiary and quaternary academic hospitals in Ontario. The hospitals varied in size (median 463 beds, range 256 to 1325). Admissions to the general internal medicine ward at each hospital occurred primarily through the emergency department as opposed to transfers from other hospitals. The majority of general internal medicine patients at each hospital were cared for by the academic teaching teams involved in this study.

Attending physicians were all certified in internal medicine through the Royal College of Physicians and Surgeons, Canada. Trainee physicians were all second (or higher) year residents in a Canadian internal medicine residency program training for the same certification. There is no mandatory requirement for specific palliative care clinical exposure in internal medicine training in Canada, although residents may participate in clinical electives to gain exposure to this area and "care of the dying" is one of the objectives of training [28].

Recruitment occurred at one vanguard site in 2014 (to hone study procedures and case reporting forms) and at the other sites between April 2017 and October 2018. All patients admitted to general internal medicine teaching teams and their corresponding physician teams were eligible for inclusion and identified on the day of survey. There were no exclusions based on patient language or cognitive status.

### Surprise question responses

Clinicians were asked "Would you be surprised if this patient died in the next 12 months?" for each of the patients for whom they were responsible, totalling two responses (one trainee physician response and one attending physician response) per patient. Trainee and attending physicians were surveyed independently with no knowledge of each others' responses. The year of medical school graduation and duration (in days) on the current service was recorded for each respondent.

### Patient data collection

Baseline patient data were collected from the hospital chart including demographic data (gender, age), baseline pre-hospital data gleaned from the clinical notes (type of residence, functional status, marital status), medical information (admitting diagnosis, comorbidities,

admission creatinine), length of stay on the day the surprise question was administered and documented cardiopulmonary resuscitation (CPR) status.

## 12-month patient mortality data collection

At every site, vital status at 12 months was recorded for each patient by reviewing the hospital electronic record or publicly available online death databases and obituaries. At all but one site, if the 12-month vital status could not be determined from the chart or death databases and obituaries, then a research assistant sent a letter to the patient's address on file offering the opportunity to opt out of a follow-up phone call. A research assistant then attempted to contact by phone all patients with undetermined vital status who did not opt out of the follow up phone call. At one site, no attempts to contact patients after discharge were permitted by the research ethics board.

## Ethics

This study was approved by the Research Ethics Boards at Sunnybrook Health Sciences Centre, St Michael's Hospital, Sinai Health Systems and University Health Network, Kingston General Hospital, and Hamilton General Hospital. The waiver for initial patient participation was important to ensure a broad enrolment population that did not exclude vulnerable subgroups such as patients with cognitive impairment, interpretation needs, or mental illness for whom the surprise question may have different operating characteristics and for whom there may be a differential risk of death or unmet palliative care needs [29].

## Statistical analysis

The primary analysis assessed agreement between attending and trainee physician 12-month surprise question responses using an adjusted Cohen's kappa measure of interrater reliability (chance-corrected agreement) to account for clustering by physician dyad [30,31]. This was calculated from the posterior probabilities of each response using a multinomial Bayesian regression model with random effects allowing for clustering by physician (**Supporting information**) [32]. Modeling the clustering by physician pair was important in order to identify the extent of variability in agreement across physician dyads [33]. The prior distributions were chosen to be minimally informative. Discordance was assessed by the relative risk of attending physicians responding "No" in discordant cases using the posterior probability distributions.

The accuracy of surprise question responses with respect to 12-month mortality was derived from 2-by-2 tables including calculation of positive and negative likelihood ratios. Bayesian multilevel models were also used to assess the prognostic characteristics of the 12-month surprise question with respect to 12-month mortality for attending physicians and trainee physicians accounting for clustering by physician dyad. This also included calculation of sensitivity, specificity, and likelihood ratios.

A further exploratory model assessed for association between kappa value and selected clinically relevant patient factors: age, sex, functional status, CPR status, admission diagnosis, and comorbidity. In these models, patients with missing data were excluded. Age was modeled with splines using 4 knots.

The Bayesian modelling program Stan was used via the statistical programming language R using the package *brms* [34–36]. Minimally informative priors were used. Models were run for 4000–7500 iterations with 1000–5000 iterations warmup, 4 chains, and 4 cores. Chains, r-hat values and parameter distributions were inspected to assess model fit. Posterior medians with 95% credible intervals were reported. Code and Monte Carlo diagnostics are available in the Supporting information.

Patients with missing surprise question response data were excluded. The impact of missing mortality data were assessed by repeating the mortality analysis with multiple imputation using 25 datasets [37].

## Results

### Demographic and clinical characteristics

We gathered surprise question responses on 546 patients from 30 attending-trainee physician dyads at 7 hospitals. The median patient age was 75 years (IQR 60–85), 260 (48%) had female sex, median length-of-stay before survey was 8 days (IQR 3–19) and most common admission diagnoses included pneumonia (10%), delirium (9%), and congestive heart failure (9%) (additional details in **Table 1**). Three patients (not included in the 546) had incomplete surprise question response data.

Trainee physicians were internal medicine residents in their second or third year of postgraduate training and the median time on service before survey was 18 days (IQR 14–21). Among attending physicians, the median time between survey and graduating medical school was 13 years (IQR 8–23) and the median time on service before survey was 10 days (IQR 9–14).

### Surprise question results

Among the 546 patients, attending physicians answered "No, I would not be surprised if this patient died in the next 12 months" ("No") for 368 patients (67%), while trainees answered "No" for 316 patients (58%). Attending and trainee physicians had the same response of "No" for 272 patients (50%) and the same response of "Yes" for 134 patients (24%) with discordant responses in the remaining 140 patients (26%) (**Table 2**).

Twelve-month surprise question responses by attending and trainee physicians showed moderate agreement with a Cohen's kappa statistic (adjusted for clustering by physician dyad) of 0.54 (95% credible interval 0.41–0.66) for the average physician dyad. The Cohen's kappa without adjustment for clustering by physician dyad was 0.46 (95% confidence interval 0.38–0.54). In discordant cases, attending physicians were more likely to answer "No" than trainee physicians (relative risk 2.36, 95% credible interval 1.22–4.98). There was moderate variation in kappa across the physician dyads (**Fig 1**). Results for the surprise question with respect to hospital discharge are available in the **Supporting information**.

### Associations between agreement and patient covariates

Associations between patient characteristics and physician responses were investigated through a larger adjusted Bayesian multinomial regression model. Two patients were excluded for missing covariate data. Age was modeled with restricted cubic splines using 4 knots and showed a nonlinear relationship between median kappa and age (**Fig 2**) with the highest agreement at approximately age 40 and the lowest agreement at approximately age 75. Agreement decreased with presence of a respiratory or cancer comorbidity and increased with presence of infection as admitting diagnosis (**Table 3**), although credible intervals were wide throughout the exploratory results.

### Comparing surprise question results and mortality

Vital status at 12 months was confirmed for 417 patients (76%). Of these patients, 160 (38%) had died. Across sites, confirmation of mortality status ranged from 47% to 99% and observed confirmed mortality rates ranged from 25% to 53%.

**Table 1. Baseline characteristics.**

| Characteristic | Number (%) |
|---|---:|
| **Hospitals** | |
| Total hospitals | 7 |
| Patients enrolled per site (median [range]) | 73 [56–104] |
| Physician dyads per hospital (median [range]) | 4 [3–6] |
| **Physicians** | |
| Total dyads | 30 |
| Patients per dyad (median [IQR]) | 18 [16–20] |
| Attending physician | |
| Years since medical graduation (median [IQR]) | 13 [8–23] |
| Days on current service (median [IQR]) | 10 [9–14] |
| Trainee physician | |
| Years since medical graduation (median [IQR]) | 2 [1–3] |
| Days on current service (median [IQR]) | 18 [14–21] |
| **Patients** | |
| Total patients | 546 (100%) |
| Patient age (years) | |
| <30 | 27 (5%) |
| 30–44 | 32 (6%) |
| 45–59 | 83 (15%) |
| 60–74 | 126 (23%) |
| 75–89 | 232 (42%) |
| ≥ 90 | 44 (8%) |
| Missing | 2 (0.4%) |
| Sex | |
| Female | 260 (48%) |
| Male | 284 (52%) |
| Missing | 2 (0.4%) |
| Residence Type (prior to hospitalization) | |
| House or Apartment | 422 (77%) |
| Retirement Home or Long-term Care | 84 (15%) |
| No fixed address | 19 (3%) |
| Other | 21 (4%) |
| Function (IADLs/ADLs) | |
| Independent for all | 179 (33%) |
| Dependent for some | 216 (40%) |
| Unknown | 151 (28%) |
| Admitting diagnoses | |
| Cardiovascular | 78 (14%) |
| Neurologic | 75 (14%) |
| Infectious | 156 (29%) |
| Acute kidney injury/metabolic abnormality | 72 (13%) |
| Other | 165 (30%) |
| Comorbidities | |
| Cardiovascular | 401 (73%) |
| Respiratory | 88 (16%) |
| Chronic kidney disease | 68 (12%) |
| Cancer | 127 (23%) |

(*Continued*)

**Table 1.** (Continued)

| Characteristic | Number (%) |
|---|---|
| CPR status | |
| Full code or not documented | 368 (67%) |
| No CPR | 178 (33%) |

Among the patients for whom mortality data were available, the probability of death given the surprise question responses was calculated using Bayesian logistic regression accounting for clustering by physician dyad. The probability of death according to surprise question responses was 57% (95% credible interval 49% to 65%) for both attending and trainee physicians responding "No" and 8.3% (95% credible interval 3.7% to 15%) for both attending and trainee physicians responding "Yes." The probability of death given a response of "No" was 49% (95% credible interval 42% to 56%) for attending physicians and 54% (95% credible interval 46% to 62%) for trainee physicians. Conversely, the probability of death given a response of "Yes" to the surprise question was 14% (95% credible interval 8% to 21%) for attending physicians and 17% (95% credible interval 11% to 23%) for trainee physicians. The analyses found very similar results after multiple imputation of missing data (**Table E1** and **Table E2** in the **Supporting information**).

The adjusted likelihood ratio for death given a response of "No" to the surprise question was 1.84 (95% credible interval 1.55 to 2.22) for trainee physicians and 1.51 (95% credible interval 1.30 to 1.72) for attending physicians. The corresponding negative likelihood ratios were 0.31 (95% credible interval 0.19 to 0.50) and 0.27 (95% confidence interval 0.13 to 0.45). Further details about sensitivity and specificity are available in **Table E3** in the **Supporting information**.

## Discussion

This study of 546 general medicine inpatients across 30 attending-trainee physician dyads of physicians showed that attending and trainee physicians had moderate agreement on the surprise question "Would you be surprised if this patient died in the next 12 months?" The adjusted Cohen's kappa was 0.54, which means that the average dyad agreed on 54% of patients after removing the patients where agreement occurred by chance [30]. The classification of this value of kappa as "moderate" is based on convention [31] and a 54% agreement rate after removing chance agreements is not reassuring in the setting of identifying patients at high risk of mortality and unmet palliative care needs.

The variation across dyads was similar in magnitude to variation by clinical characteristics in exploratory analyses. Agreement by Cohen's kappa in this study was similar to that seen in

**Table 2. 12-month surprise question responses of attending and trainee physicians.**

| Trainee Physician | | Attending Physician | | |
|---|---|---|---|---|
| | | "No, I would not be surprised if this patient died in the next 12 months." | "Yes, I would be surprised if this patient died in the next 12 months." | |
| | "No, I would not be surprised if this patient died in the next 12 months." | 272 (49%) | 44 (8%) | 316 (57%) |
| | "Yes, I would be surprised if this patient died in the next 12 months." | 96 (18%) | 134 (25%) | 230 (43%) |
| | | 368 (67%) | 178 (33%) | 546 (100%) |

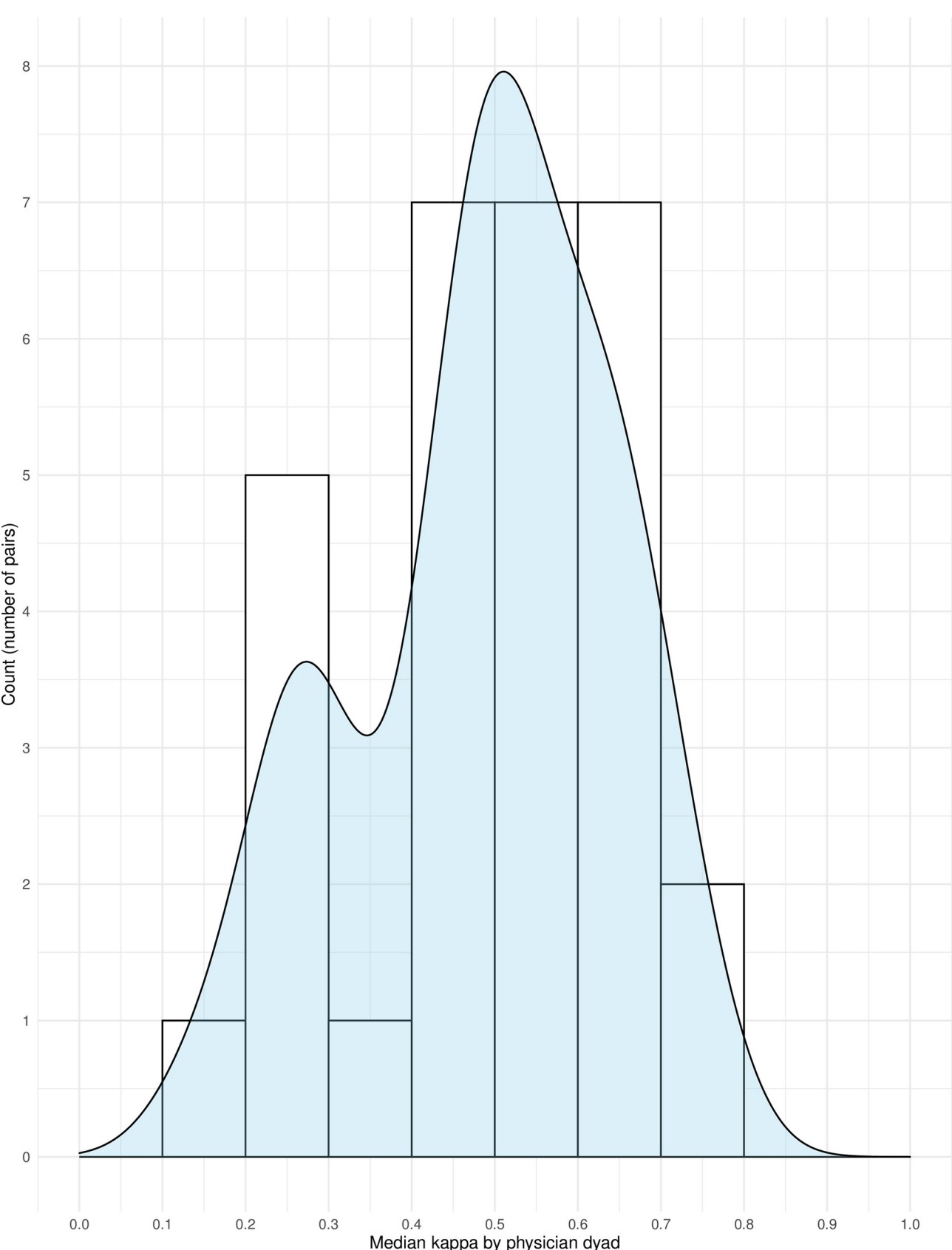

**Fig 1. Histogram and density plot of median kappa values by physician dyad.** This figure shows the distribution of kappa values according to physician dyad by histogram (bars) with a density plot overlay (light blue). Most clinician dyads had a kappa value above 0.4, but some outliers had kappa values between 0.1 and 0.3.

another study [15] comparing predictions of nurses and physicians, suggesting that moderate agreement on prognosis may be common to other dyads of clinicians beyond trainee-attending dyads. Taken together, these findings imply that discussion of patient prognosis between multidisciplinary clinical team members is essential to ensure appropriate engagement of palliative care services and to avoid mixed messages to patients and family members.

The surprise question may be useful to "rule out" a high risk of mortality, but it is not sufficient as a standalone screening measure for identifying patients who have high risk of mortality. A response of "Yes" on the surprise question was associated with low 12-month mortality in this and other settings, but the sensitivity and specificity remain inconsistent and unsuited to a screening tool [2,38]. The lowest 12-month mortality rate was seen in patients where both attending and trainee physician responded "Yes" to the surprise question, similar to other studies where combining predictions of multiple clinical team members yielded the best predictions [15]. In contrast to previous studies investigating physician-estimated prognoses which found correlation between accuracy and level of training [11,39–42], attending physician predictions were not more accurate than trainee physician predictions with respect to mortality in our study. This could mean that both attending and trainee physicians would benefit from educational interventions focused on prognosis. Alternatively, accurate prognostication may require clinicians to combine clinical insights with novel tools not yet in clinical use, such as automated screening tools derived from electronic medical records [43–46].

This study has strengths including a pragmatic approach, inclusive enrollment criteria, paired design, multicenter data, and statistical methods that account for clustering. The simplicity of the surprise question allowed us to include all patients within our inclusion criteria with no planned or unplanned systematic exclusions on the day of survey. Our paired design minimized patient-level confounding and comparing to the most senior trainee as comparator minimized error due to medical inexperience. The choice of Bayesian modeling permitted a more sophisticated analysis of the Cohen's kappa coefficient which has not previously been used in analyses of the surprise question [2,38].

The main limitation of this study is the lack of data on unmet palliative care needs such as uncontrolled pain or nausea, psychological distress about the dying process, or ignorance about the available options for end-of-life care. This limitation is also present in other research on the surprise question [22]. It is unknown whether the attending or trainee physician is more likely to be correct with respect to palliative care needs when there is discordance. A related limitation is a lack of corresponding qualitative data including information about the reasoning behind the responses of each participant. It is also unknown whether the two physicians were aware when they did not agree, or if this was discussed for any of the patients in the study.

A more fundamental limitation is that the surprise question is a subjective instrument that integrates a healthcare practitioner's expertise, knowledge, and personal biases [47]. This study includes uncertainty both from the intrinsic uncertainty in estimating a patients' prognosis and from the uncertainty in how clinicians interpret this uncertainty in responding to the surprise question. The validity of the surprise question with respect to its intended purpose of identifying patients with palliative care needs remains unclear. Even if the surprise question reliably identifies those patients, identification alone is necessary but not sufficient for meeting those needs [48,49]. Future research may need to focus on identification of unmet palliative

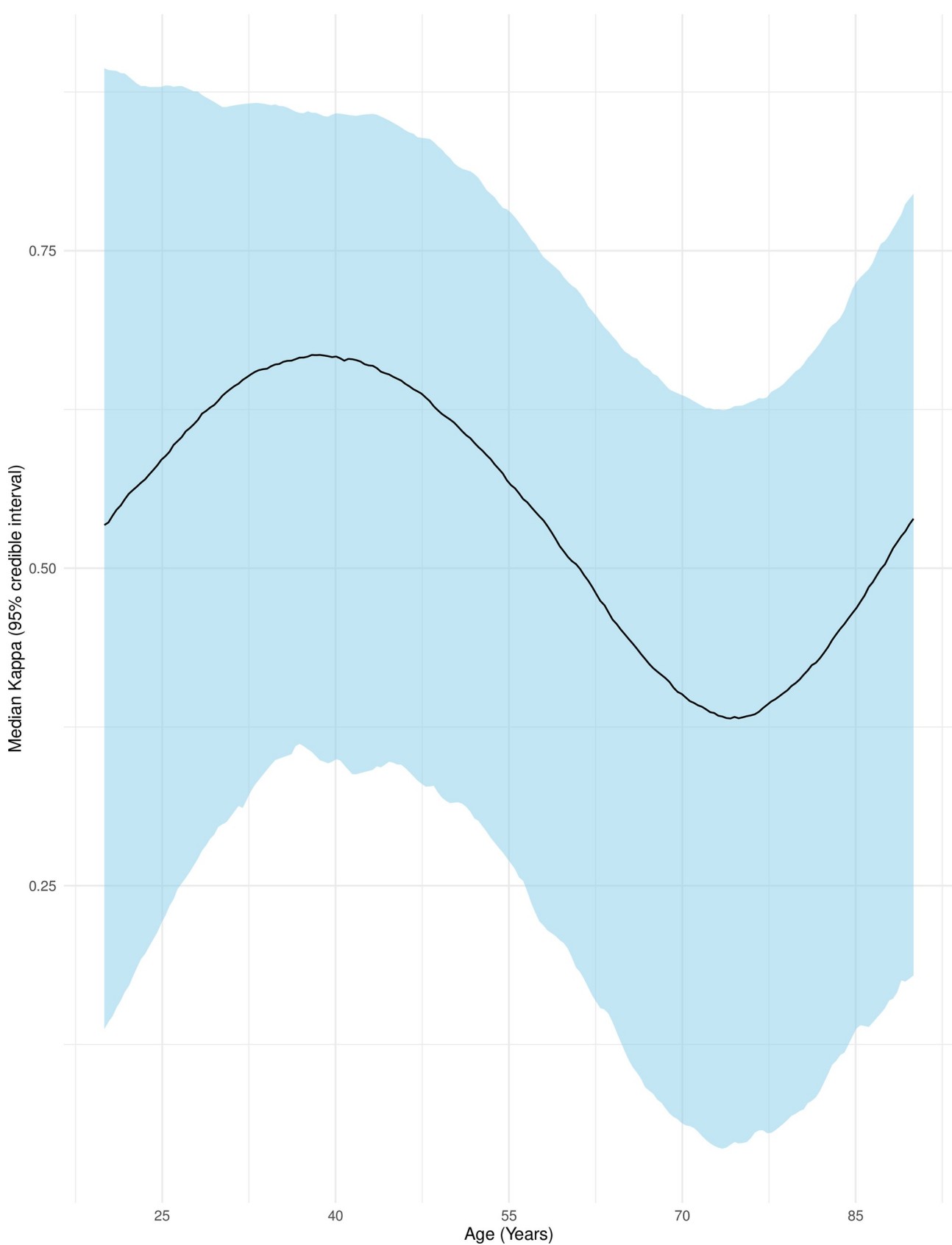

**Fig 2. Agreement between attending and trainee physician surprise question responses by patient age.** This figure shows the median Cohen's kappa (adjusted for clustering by physician dyad) as a black line with 95% credible intervals as surrounding blue ribbon. Age was modeled with restricted cubic splines using 4 knots. Other variables were set to: Female, independent, full code, infection as admitting diagnosis, and presence of a cardiovascular comorbidity only.

care needs as opposed to predicting mortality, as these needs may be a better marker of potential benefit from a palliative approach and provide tangible targets for the clinical team.

## Conclusion

Using the surprise question to measure prognosis, the average general internal medicine attending and trainee physician agreed on the prognosis of only 54% of their patients after correcting for chance agreement. These data remind clinicians of the subjectivity in formulating patient prognoses, and the importance of routine discussion of patient prognosis between team members in order to ensure a coherent clinical plan including clear communication with patients and appropriate engagement of palliative care services.

**Table 3. Agreement between attending and trainee physician by patient subgroup, adjusted for multiple clinical covariates.**

| Patient characteristic | Median Kappa (95% credible interval) |
| --- | --- |
| Age (years) | |
| 30 | 0.69 (0.34 to 0.89) |
| 45 | 0.72 (0.41 to 0.90) |
| 60 | 0.59 (0.27 to 0.82) |
| 75 | 0.47 (0.14 to 0.74) |
| 90 | 0.64 (0.31 to 0.87) |
| Sex | |
| Female | 0.59 (0.26 to 0.81) |
| Male | 0.61 (0.35 to 0.84) |
| Function (IADLs/ADLs) | |
| Independent for all | 0.59 (0.28 to 0.83) |
| Dependent for some | 0.55 (0.19 to 0.82) |
| Unknown | 0.56 (0.16 to 0.80) |
| CPR status | |
| Full code and not documented | 0.57 (0.28 to 0.83) |
| Not for CPR | 0.46 (0.02 to 0.78) |
| Admitting diagnosis | |
| Cardiovascular | 0.59 (0.29 to 0.81) |
| Neurologic | 0.50 (0.21 to 0.73) |
| Infectious | 0.71 (0.42 to 0.87) |
| AKI/metabolic | 0.41 (-0.07 to 0.75) |
| Other | 0.31 (-0.12 to 0.67) |
| Comorbidities | |
| Cardiovascular | 0.46 (0.06 to 0.80) |
| Respiratory | 0.17 (-0.20 to 0.64) |
| Chronic kidney disease | 0.42 (0.00 to 0.82) |
| Cancer | 0.29 (-0.04 to 0.75) |

Unless otherwise noted, the baseline characteristics used for calculating median kappas and 95% credible intervals were: Age 60 years, female, independent, full code, cardiovascular admitting diagnosis and cardiovascular comorbidity.

## Supporting information

**S1 File.**
(DOCX)

**S2 File.**
(DOCX)

**S3 File.**
(CSV)

## Acknowledgments

An appreciative thanks to the following people: Carol Mantle and Marilyn Swinton for assistance with data collection and project coordination at Hamilton Health Sciences. Katherine Allan for assistance with data collection and project coordination at St Michael's Hospital. Ellen Koo for assistance with data collection and project coordination at Toronto General and Toronto Western Hospitals. Nicole Marinoff for assistance with project coordination at Sunnybrook Health Sciences Center. Julia Kruizinga for assistance with data collection at Kingston Health Science Centre.

## Author Contributions

**Conceptualization:** Christopher J. Yarnell, Laura M. Jewell, Ruxandra Pinto, Luke A. Devine, James Downar, Roy Ilan, Shail Rawal, Natalie Wong, John J. You, Rob A. Fowler.

**Data curation:** Christopher J. Yarnell, Laura M. Jewell, Alex Astell, Luke A. Devine, James Downar, Roy Ilan, Shail Rawal, Natalie Wong, John J. You, Rob A. Fowler.

**Formal analysis:** Christopher J. Yarnell, Laura M. Jewell, Ruxandra Pinto, Michael E. Detsky, Natalie Wong, John J. You, Rob A. Fowler.

**Funding acquisition:** Christopher J. Yarnell, Laura M. Jewell, Rob A. Fowler.

**Investigation:** Christopher J. Yarnell, Laura M. Jewell, Alex Astell, James Downar, Roy Ilan, Shail Rawal, Natalie Wong, John J. You, Rob A. Fowler.

**Methodology:** Christopher J. Yarnell, Laura M. Jewell, Alex Astell, Ruxandra Pinto, James Downar, Roy Ilan, Shail Rawal, Natalie Wong, John J. You, Rob A. Fowler.

**Project administration:** Christopher J. Yarnell, Laura M. Jewell, Alex Astell, Luke A. Devine, James Downar, Roy Ilan, Shail Rawal, Natalie Wong, John J. You, Rob A. Fowler.

**Resources:** Christopher J. Yarnell, Laura M. Jewell, Alex Astell, Luke A. Devine, James Downar, Roy Ilan, Shail Rawal, Natalie Wong, John J. You, Rob A. Fowler.

**Software:** Christopher J. Yarnell.

**Supervision:** Rob A. Fowler.

**Visualization:** Christopher J. Yarnell, Rob A. Fowler.

**Writing – original draft:** Christopher J. Yarnell.

**Writing – review & editing:** Christopher J. Yarnell, Laura M. Jewell, Alex Astell, Ruxandra Pinto, Luke A. Devine, Michael E. Detsky, James Downar, Roy Ilan, Shail Rawal, Natalie Wong, John J. You, Rob A. Fowler.

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
