## [Decision Letter · Decision Letter 0]

1 Dec 2020

PONE-D-20-25533

Observational study of agreement between attending and trainee physicians on the surprise question: “Would you be surprised if this patient died in the next 12 months?”

PLOS ONE

Dear Dr. Yarnell,

Thank you for submitting your manuscript to PLOS ONE. After careful consideration, we feel that it has merit but does not fully meet PLOS ONE’s publication criteria as it currently stands. Therefore, we invite you to submit a revised version of the manuscript that addresses the points raised during the review process.

ACADEMIC EDITOR: Minor revision is required for the current manuscript. Please respond to the questions from the reviewers.

We look forward to receiving your revised manuscript.

Kind regards,

Jason Chia-Hsun Hsieh, M.D. Ph.D

Academic Editor

PLOS ONE

Journal Requirements:

Minor revision is required for the current manuscript. Please respond to the questions from the reviewers.

Reviewers' comments:

Reviewer's Responses to Questions

**Comments to the Author**

1. Is the manuscript technically sound, and do the data support the conclusions?

Reviewer #1: Yes

Reviewer #2: Yes

2. Has the statistical analysis been performed appropriately and rigorously? 

Reviewer #1: Yes

Reviewer #2: Yes

3. Have the authors made all data underlying the findings in their manuscript fully available?

Reviewer #1: Yes

Reviewer #2: No

4. Is the manuscript presented in an intelligible fashion and written in standard English?

Reviewer #1: Yes

Reviewer #2: Yes

5. Review Comments to the Author

Reviewer #1: Thank you for sharing with me this fascinating paper to review that chimes closely with my clinical and research interests in a recent cluster trail of an intervention to better serve patients with multiple morbidities whose situations are clinically uncertain and where they were considered to be at risk of dying during their hospital say despite receiving treatment. We experienced innumerable problems in practically operationalising this issue [1].

My question as to why it's so important to be able to predict which patients are at risk of dying in the next 12 months is as present now as it was then. Perhaps, greater emphasis of its utility is required in the opening paragraphs. If we are to focus on patient need (physical and well as psychosocial issues) and need alone i.e. the ability or capacity to benefit for clinical intervention that is seen to be effective, where is the case for essentially tossing a coin? My point here is that prognostication is a contested field and I think this needs to be unpicked a bit more critically in the opening paragraphs. My view is prognostic models vary in levels of sophistication, ranging from clinical intuition to more intricate multivariate statistical models that combine multiple factors to yield an assessment [2]. We discovered that where ‘risk of dying’ is focused on there is a risk of two sampling biases being present due to the unknown (or the inconsistent) manner in which health professionals currently interpret risk – the question you are addressing in this study albeit between different levels of seniority Firstly, the unpredictable and often unreliable identification of potential cases and secondly their exclusion. Another concern I have is that although models to enhance the identification of dying patients using prognostic models are improving for patients with cancer [3] [4] [5] (a lot of this literature is indeed referred to), there is far less consensus on methods to assess patients with non-malignant conditions, which are more common in many health care settings studies [6, 7]. Forgive my rant but these are indeed important areas to focus on at some point in this paper.

I like the aim of this study, however, t I would be grateful if the authors would justify why the comparison between attending and trainee physicians? Intuitively I think I know why but it would be useful to shed light on any underlying hypotheses the authors wish to examine. For example, does the ability and confidence in prognostication get better with experience

The methods are described elegantly. I like the term vanguard site. We often refer to this as a pilot site, but I can live with that. It would be useful to have a little more contextual information about the hospitals concerning the populations they serve and how many patients they typically see each year, and at the very least how many patients the wards serve.

A question. Are you able to shed light on the education that the trainees might have previously received about assessing risk and prognostication? Is this information available from their curriculum?

The 12-month patient mortality data collection section of this paper is important. Were there any ethical issues raised by contacting the patient’s address on file offering the opportunity to opt-out of a follow-up phone call? This is not alluded to with the subsequent phone call.

Good to me that vulnerable group were also considered to be important to include in this study. Vulnerable groups are often excluded from study and palliative care study in particular. May be worth citing a reference or two to support this section. Kipnis among others have written about this is, as have I historically.

A clear statistical plan presented clearly and elegantly. Very replicable and transparent.

Results.

An ‘n’ for table 1 would be useful to include

Page 12, it appears no information was collected regarding training in prognostication

Table 2 works well. Could also be presented in the form of a figure?

Useful supplement files. Thank you.

It perhaps would have been useful to shed light why its important to understand the possible relationship between levels of agreement and co-variates. Perhaps this goes back to the original aim of the study and hypotheses that could have been set to test subsequently.

Discussion

Page 17, line 328-329. I agree with this statement that clinical team members need to communicate with each to better effect to identify patients where there is a risk of dying in the next 12mopnths. I take it this would include other members of the multi-professional team, not just physicians. It still begs the question of whether the focus should be on the needs of patients irrespective of prognosis. This is linked to the limitations of this study where the authors state there was a lack of data on the unmet palliative care needs of patients. What do they mean by this?

I agree that prognostication offers a pragmatic solution to identify a group of patients who might benefit from specialist palliative and end of life care. However, given that this approach relies heavily on subjectivity (the authors acknowledge this) in prognostication perhaps it should be just plain avoided? Instead, greater emphasis is placed on objective clinical indicators, for example, poor performance status scores, the presence and severity of cognitive impairment, weight loss, and dysphagia, and of course the presence of distressing symptoms that can be ameliorated.

1. Koffman, J., et al., Managing uncertain recovery for patients nearing the end of life in hospital: a mixed-methods feasibility cluster randomised controlled trial of the AMBER care bundle. Trials, 2019. 20(1): p. 506.

2. Yourman, L.C., et al., Prognostic indices for older adults: a systematic review. Jama, 2012. 307(2): p. 182-92.

3. Stone, P. and S. Lund, Predicting prognosis in patients with advanced cancer. Annals of Oncology, 2006. 18(6): p. 971-976.

4. Stone, P., et al., Patients' reports or clinicians' assessments: which are better for prognosticating? BMJ Supportive & Palliative Care, 2012. 2(3): p. 219-223.

5. White, N., et al., A systematic review of predictions of survival in palliative care: How accurate are clinicians and who are the experts? PLoS ONE, 2016. 11(8): p. e0161407.

6. Nutter, A.L., T. Tanawuttiwat, and M.A. Silver, Evaluation of 6 Prognostic Models Used to Calculate Mortality Rates in Elderly Heart Failure Patients With a Fatal Heart Failure Admission. Congestive Heart Failure, 2010. 16(5): p. 196-201.

7. Glimelius, B., Palliative medicine ? A research challenge Acta Oncologica, 2000. 39(8): p. 891-893.

Reviewer #2: An interesting and well written report. I do have some comments.

Abstract, page 4 line 114 – where it reads “We undertook to determine agreement…” should it read “We undertook this study to determine agreement…”

Methods, page 8 line 191 – how was this done exactly? Were attendings and trainees in the same room at the same time? Were they asked to reply verbally or in writing? If this is the case could that not allow for biases, as trainees might be more prone to agree, or fear to disagree with their attendings? This would be a limitation of the study

Results, page 13, lines 249-252: it would have been important to know and to state in the paper if any of the participating physicians had palliative care training (either basic or advance) given that, identifying patients with palliative care needs and patients at the end of their disease trajectory might be done more easily after being exposed to that specific training, as would be to make the case for answering “no” or “yes” to the surprise question. Not knowing about specific palliative care training can be a potential limitation of this study. Additionally, it could also be potentially used to make the point authors raise in the discussion, page 17, lines 331,332 “The surprise question may be useful to “rule out” a high risk of mortality, but it is not sufficient as a standalone screening measure for identifying patients who have high risk of mortality.”; lines 358, 359 “Even if the surprise question reliably identifies those patients, identification alone is necessary but not sufficient for meeting those needs” and in the conclusion, lines 365, 366 “appropriate engagement of palliative care services.”

Conclusion: although I agree with authors’ conclusion, there is a vital component missing, which I feel should be mentioned in this section, which is the importance of palliative care training for physicians working in these services.

Typos: “data was” instead of “data were” occurs throughout the paper. Please amend this.

6. PLOS authors have the option to publish the peer review history of their article (what does this mean?). If published, this will include your full peer review and any attached files.

Reviewer #1: **Yes: **Jonathan Koffman

Reviewer #2: No

---

## [Author Response · Author response to Decision Letter 0]

24 Dec 2020

Reviewer 1

REVIEWER: Thank you for sharing with me this fascinating paper to review that chimes closely with my clinical and research interests in a recent cluster trail of an intervention to better serve patients with multiple morbidities whose situations are clinically uncertain and where they were considered to be at risk of dying during their hospital say despite receiving treatment. We experienced innumerable problems in practically operationalising this issue [1].

RESPONSE: Thank you for sharing your experience. We have added this work as a citation to support the negative impact of uncertainty on patients, physicians, and communication between the two. Citation was added at the end of the following new sentence: “Uncertainty in patient prognosis can increase patient distress, impair communication between physicians and patients, and delay appropriate adoption of a palliative approach.”

REVIEWER: My question as to why it's so important to be able to predict which patients are at risk of dying in the next 12 months is as present now as it was then. Perhaps, greater emphasis of its utility is required in the opening paragraphs. 

RESPONSE: We have updated the Intro paragraph 3 to greater emphasise the utility of identifying which patients are at risk of dying in the next 12 months. We added: “Improving the identification of patients at high risk of dying can improve clinical care through providing more certainty to patients, more clarity for clinicians making recommendations about invasive interventions or investigations, and triggering earlier appropriate activation of palliative care services.”

REVIEWER: If we are to focus on patient need (physical and well as psychosocial issues) and need alone i.e. the ability or capacity to benefit for clinical intervention that is seen to be effective, where is the case for essentially tossing a coin? My point here is that prognostication is a contested field and I think this needs to be unpicked a bit more critically in the opening paragraphs. 

RESPONSE: We agree that prognostication is difficult, nuanced, unlikely to be fully captured by a binary indicator, and that a patient’s potential benefit from involvement of palliative care or consideration of end-of-life planning is not solely determined by their prognosis in terms of time. We have added the following sentence to the opening paragraph to unpack these complex notions: “Further, the risk of death may not correlate with other important aspects of patient care such as uncontrolled symptoms or psychological distress.”

REVIEWER: My view is prognostic models vary in levels of sophistication, ranging from clinical intuition to more intricate multivariate statistical models that combine multiple factors to yield an assessment [2]. We discovered that where ‘risk of dying’ is focused on there is a risk of two sampling biases being present due to the unknown (or the inconsistent) manner in which health professionals currently interpret risk – the question you are addressing in this study albeit between different levels of seniority Firstly, the unpredictable and often unreliable identification of potential cases and secondly their exclusion. 

RESPONSE: We agree that one consequence of the experimental design of this project is that we incorporate error both at the level of clinicians making an accurate prediction and at the level of clinicians interpreting how to respond to the surprise question given their prediction of risk. We have tried to emphasize this further in the discussion section.

REVIEWER: Another concern I have is that although models to enhance the identification of dying patients using prognostic models are improving for patients with cancer [3] [4] [5] (a lot of this literature is indeed referred to), there is far less consensus on methods to assess patients with non-malignant conditions, which are more common in many health care settings studies [6, 7]. 

RESPONSE: The inclusion criteria for this study were broad to ensure the information is applicable to patients with both cancer and non-cancer diagnoses. We have highlighted this point in the discussion with the following sentence added to the final limitations paragraph: “This study includes uncertainty both from the intrinsic uncertainty in estimating a patients’ prognosis and from the uncertainty in how clinicians interpret this uncertainty in responding to the surprise question.”

REVIEWER: I like the aim of this study, however, t I would be grateful if the authors would justify why the comparison between attending and trainee physicians? Intuitively I think I know why but it would be useful to shed light on any underlying hypotheses the authors wish to examine. For example, does the ability and confidence in prognostication get better with experience.

RESPONSE: Thank you for the feedback. We have endeavoured to better justify the rationale for comparison between trainee and attending physicians in the introduction section. The second paragraph of the introduction has been rewritten as follows:

“One way to gain insight into clinician estimates of prognosis is to investigate how prognostic accuracy varies with training by comparing the prognostic estimates of attending and trainee physicians. Studying differences in prognostic estimates between trainee and attending physicians provides a way to investigate the development of prognostic skills, including whether or not accuracy improves with clinical training. Prognostic discordance and factors associated with discordance may identify an opportunity for education, and may highlight the patients for whom prognosis is most difficult. Discordance between trainee and attending physician prognoses also has practical implications because trainee physicians play an important role for inpatients at academic health centres, where discordance may cause confusion in clinical plans, mixed messages for patients, and may partially explain observed differences between received and documented patient end-of-life care preferences. The prevalence of discordance between attending and trainee physician prognoses is unknown.”

REVIEWER: The methods are described elegantly. I like the term vanguard site. We often refer to this as a pilot site, but I can live with that. It would be useful to have a little more contextual information about the hospitals concerning the populations they serve and how many patients they typically see each year, and at the very least how many patients the wards serve.

RESPONSE: We have added contextual information about the hospitals to the methods section. The first paragraph of the Study population subsection now reads: 

“This was a prospective cohort study of general internal medicine inpatients and their corresponding physicians at 7 tertiary and quaternary academic hospitals in Ontario. The hospitals varied in size (median 463 beds, range 256 to 1325). Admissions to the general internal medicine ward at each hospital occurred primarily through the emergency department as opposed to transfers from other hospitals. The majority of general internal medicine patients at each hospital were cared for by the academic teaching teams involved in this study.”

REVIEWER: A question. Are you able to shed light on the education that the trainees might have previously received about assessing risk and prognostication? Is this information available from their curriculum?

RESPONSE: We have added information about the training backgrounds of the attending and trainee physicians:

“Attending physicians were all certified in internal medicine through the Royal College of Physicians and Surgeons, Canada. Trainee physicians were all second (or higher) year residents in a Canadian internal medicine residency program training for the same certification. There is no mandatory requirement for specific palliative care clinical exposure in internal medicine training in Canada, although residents may participate in clinical electives to gain exposure to this area and “care of the dying” is one of the objectives of training.” 

REVIEWER: The 12-month patient mortality data collection section of this paper is important. Were there any ethical issues raised by contacting the patient’s address on file offering the opportunity to opt-out of a follow-up phone call? This is not alluded to with the subsequent phone call.

RESPONSE: We did not encounter any issues during the phone calls. However, as mentioned in the methods, one site did not allow any contact of patients with outcomes not apparent in the hospital chart. The research ethics board for that site felt that this would only be possible if patients had provided written, informed consent to a follow up phone call during the index admission. We did not opt to go that route for both scientific reasons (this would potentially exclude many vulnerable patients with cognitive impairment or non-English first languages) and logistical reasons (lacking funding to pursue such a larger project). We have updated the last sentence in the subsection 12-month patient mortality data collection to read: “At one site, no attempts to contact patients after discharge were permitted by the research ethics board.”

REVIEWER: Good to me that vulnerable group were also considered to be important to include in this study. Vulnerable groups are often excluded from study and palliative care study in particular. May be worth citing a reference or two to support this section. Kipnis among others have written about this is, as have I historically.

RESPONSE: Thank you for the suggestion. I enjoyed reading your 2009 paper on this topic and I’m glad you pointed it out. We have added it as a citation to our subsection entitled “Ethics.”

REVIEWER: A clear statistical plan presented clearly and elegantly. Very replicable and transparent.

RESPONSE: Thank you

REVIEWER: An ‘n’ for table 1 would be useful to include

RESPONSE: Done

REVIEWER: Page 12, it appears no information was collected regarding training in prognostication

RESPONSE: No formal information was collected. All participating trainees were internal medicine residents training for Royal College of Physicians and Surgeons Canada certification.

REVIEWER: Table 2 works well. Could also be presented in the form of a figure?

RESPONSE: Thank you. For now, we have left this in the classic “2 by 2 table” format, but if the editors prefer this be transformed to a bar graph we are happy to attempt it.

REVIEWER: It perhaps would have been useful to shed light why its important to understand the possible relationship between levels of agreement and co-variates. Perhaps this goes back to the original aim of the study and hypotheses that could have been set to test subsequently.

RESPONSE: We have added text to the methods and results section to highlight the usefulness of performing a multilevel analysis of this clustered data.

REVIEWER: Page 17, line 328-329. I agree with this statement that clinical team members need to communicate with each to better effect to identify patients where there is a risk of dying in the next 12mopnths. I take it this would include other members of the multi-professional team, not just physicians. 

RESPONSE: We agree and have clarified the statement to ensure that the importance of communication with the multi-professional team is evident.

REVIEWER: It still begs the question of whether the focus should be on the needs of patients irrespective of prognosis. This is linked to the limitations of this study where the authors state there was a lack of data on the unmet palliative care needs of patients. What do they mean by this?

RESPONSE: We agree that perhaps a different approach could have been to ask physicians which of their patients they felt were likely to have unmet palliative care needs, such as uncontrolled pain or nausea, uncertainty and anxiety about end-of-life care, or a desire to discuss and alleviate deeply held fears about the dying process. The process of conducting a project focused on prognosticating mortality has demonstrated to us that this approach does not necessarily capture unmet patient needs, which may be a better marker of who would benefit from increased involvement with palliative care resources. We have added these concepts to the discussion section.

REVIEWER: I agree that prognostication offers a pragmatic solution to identify a group of patients who might benefit from specialist palliative and end of life care. However, given that this approach relies heavily on subjectivity (the authors acknowledge this) in prognostication perhaps it should be just plain avoided? Instead, greater emphasis is placed on objective clinical indicators, for example, poor performance status scores, the presence and severity of cognitive impairment, weight loss, and dysphagia, and of course the presence of distressing symptoms that can be ameliorated.

RESPONSE: We agree, and we have adjusted the discussion section to try and better capture this point.

1. Koffman, J., et al., Managing uncertain recovery for patients nearing the end of life in hospital: a mixed-methods feasibility cluster randomised controlled trial of the AMBER care bundle. Trials, 2019. 20(1): p. 506.

4. Stone, P., et al., Patients' reports or clinicians' assessments: which are better for prognosticating? BMJ Supportive & Palliative Care, 2012. 2(3): p. 219-223.

5. White, N., et al., A systematic review of predictions of survival in palliative care: How accurate are clinicians and who are the experts? PLoS ONE, 2016. 11(8): p. e0161407.

6. Nutter, A.L., T. Tanawuttiwat, and M.A. Silver, Evaluation of 6 Prognostic Models Used to Calculate Mortality Rates in Elderly Heart Failure Patients With a Fatal Heart Failure Admission. Congestive Heart Failure, 2010. 16(5): p. 196-201.

7. Glimelius, B., Palliative medicine ? A research challenge Acta Oncologica, 2000. 39(8): p. 891-893.

Reviewer 2

REVIEWER: Abstract, page 4 line 114 – where it reads “We undertook to determine agreement…” should it read “We undertook this study to determine agreement…”

RESPONSE: We have adjusted the abstract.

REVIEWER: Methods, page 8 line 191 – how was this done exactly? Were attendings and trainees in the same room at the same time? Were they asked to reply verbally or in writing? If this is the case could that not allow for biases, as trainees might be more prone to agree, or fear to disagree with their attendings? This would be a limitation of the study

RESPONSE: We have described the process in more detail. The trainee and attending physicians were not aware of each others’ responses.

REVIEWER: Results, page 13, lines 249-252: it would have been important to know and to state in the paper if any of the participating physicians had palliative care training (either basic or advance) given that, identifying patients with palliative care needs and patients at the end of their disease trajectory might be done more easily after being exposed to that specific training, as would be to make the case for answering “no” or “yes” to the surprise question. Not knowing about specific palliative care training can be a potential limitation of this study. Additionally, it could also be potentially used to make the point authors raise in the discussion, page 17, lines 331,332 “The surprise question may be useful to “rule out” a high risk of mortality, but it is not sufficient as a standalone screening measure for identifying patients who have high risk of mortality.”; lines 358, 359 “Even if the surprise question reliably identifies those patients, identification alone is necessary but not sufficient for meeting those needs” and in the conclusion, lines 365, 366 “appropriate engagement of palliative care services.”

RESPONSE: We have added details about palliative care training to the methods section of the paper.

REVIEWER: Conclusion: although I agree with authors’ conclusion, there is a vital component missing, which I feel should be mentioned in this section, which is the importance of palliative care training for physicians working in these services.

RESPONSE: Thank you. We have reinforced the importance of palliative care training for ensuring high-quality end-of-life care.

REVIEWER: Typos: “data was” instead of “data were” occurs throughout the paper. Please amend this.

RESPONSE: Thank you. We have amended this.

---

## [Decision Letter · Decision Letter 1]

10 Feb 2021

Observational study of agreement between attending and trainee physicians on the surprise question: “Would you be surprised if this patient died in the next 12 months?”

PONE-D-20-25533R1

Dear Dr. Yarnell,

We’re pleased to inform you that your manuscript has been judged scientifically suitable for publication and will be formally accepted for publication once it meets all outstanding technical requirements.

Kind regards,

Jason Chia-Hsun Hsieh, M.D. Ph.D

Academic Editor

PLOS ONE

Additional Editor Comments (optional):

The questions were answered adequately.

Reviewers' comments:

Reviewer's Responses to Questions

**Comments to the Author**

1. If the authors have adequately addressed your comments raised in a previous round of review and you feel that this manuscript is now acceptable for publication, you may indicate that here to bypass the “Comments to the Author” section, enter your conflict of interest statement in the “Confidential to Editor” section, and submit your "Accept" recommendation.

Reviewer #1: All comments have been addressed

Reviewer #2: All comments have been addressed

2. Is the manuscript technically sound, and do the data support the conclusions?

Reviewer #1: Yes

Reviewer #2: Yes

3. Has the statistical analysis been performed appropriately and rigorously? 

Reviewer #1: Yes

Reviewer #2: Yes

4. Have the authors made all data underlying the findings in their manuscript fully available?

Reviewer #1: Yes

Reviewer #2: Yes

5. Is the manuscript presented in an intelligible fashion and written in standard English?

Reviewer #1: Yes

Reviewer #2: Yes

6. Review Comments to the Author

Reviewer #1: Thank you for taking the time to read through my comments and co-reviewer for that matter. I believe you have now adequately addressed all the issues I raised.

Reviewer #2: All points raised were addressed by authors. The paper is improved. An extremely well written report. Useful for clinical practice and for research. I have no further comments to add.

7. PLOS authors have the option to publish the peer review history of their article (what does this mean?). If published, this will include your full peer review and any attached files.

Reviewer #1: **Yes: **Jonathan S Koffman

Reviewer #2: No

---

## [Editor Report · Acceptance letter]

11 Feb 2021

PONE-D-20-25533R1 

Observational study of agreement between attending and trainee physicians on the surprise question: “Would you be surprised if this patient died in the next 12 months?” 

Dear Dr. Yarnell:

I'm pleased to inform you that your manuscript has been deemed suitable for publication in PLOS ONE. Congratulations! Your manuscript is now with our production department. 

Kind regards, 

on behalf of

Dr. Jason Chia-Hsun Hsieh 

Academic Editor

PLOS ONE